# FAST UNSUPERVISED DEEP OUTLIER MODEL SELECTION WITH HYPERNETWORKS

## ABSTRACT

Outlier detection (OD) has a large literature as it finds many applications in the real world. Deep neural network based OD (DOD) has seen a recent surge of attention thanks to the many advances in deep learning. In this paper, we consider a critical-yet-understudied challenge with unsupervised DOD, that is, effective hyperparameter (HP) tuning or model selection. While prior work report the sensitivity of OD models to HP choices, it is ever so critical for the modern DOD models that exhibit a long list of HPs. We introduce HYPER for HP-tuning DOD models, tackling two key challenges: (1) validation without supervision (due to lack of labeled outliers), and (2) efficient search of the HP/model space (due to exponential growth in the number of HPs). A key idea is to design and train a novel hypernetwork (HN) that maps HPs onto optimal weights of the main DOD model. In turn, HYPER capitalizes on a *single* HN that can dynamically generate weights for *many* DOD models (corresponding to varying HPs), which offers significant speed-up. In addition, it employs meta-learning on historical OD tasks with labels to train a performance estimator function, likewise trained with our proposed HN efficiently. Extensive experiments on a testbed of 35 benchmark datasets show that HYPER achieves 7% performance improvement and $4.2\times$ speed up over the latest baseline, establishing the new state-of-the-art.

## 1 INTRODUCTION

With recent advances in deep learning, deep neural network (NN) based outlier detection (DOD) has seen a surge of attention Pang et al. (2021); Ruff et al. (2021). These models, however, inherit many hyperparameters (HPs); architectural (e.g. depth, width), regularization (e.g. dropout rate, weight decay), and optimization HPs (e.g. learning rate). As expected, their performance is highly sensitive to the HP settings Ding et al. (2022). This makes effective HP or model selection critical, yet computationally costly as the model space gets exponentially large in the number of HPs.

Hyperparameter optimization (HPO) can be written as a bi-level problem, where the optimal parameters $\mathbf{W}^*$ (i.e. NN weights) on the training set depend on the hyperparameters $\boldsymbol{\lambda}$.

$$\boldsymbol{\lambda}^* = \arg\min_{\boldsymbol{\lambda}} \ \mathcal{L}_{\text{val}}(\boldsymbol{\lambda}; \mathbf{W}^*) \quad s.t. \quad \mathbf{W}^* = \arg\min_{\mathbf{W}} \ \mathcal{L}_{\text{trn}}(\mathbf{W}; \boldsymbol{\lambda}) \tag{1}$$

where $\mathcal{L}_{\text{val}}$ and $\mathcal{L}_{\text{trn}}$ denote the validation and training losses, respectively. There is a body of literature on HPO for supervised settings Bergstra & Bengio (2012); Li et al. (2017); Shahriari et al. (2016), and several for supervised OD that use labeled outliers for validation Li et al. (2021; 2020); Lai et al. (2021). While supervised model selection leverages $\mathcal{L}_{\text{val}}$, *unsupervised* OD posits a unique challenge: it does not exhibit labeled hold-out data to evaluate $\mathcal{L}_{\text{val}}$. It is unreliable to employ the same $\mathcal{L}_{\text{trn}}$ loss as $\mathcal{L}_{\text{val}}$ as models with minimum training loss do not necessarily associate with accurate detection Ding et al. (2022) (e.g. autoencoder with low reconstruction error has likely missed the outliers).

### 1.1 RELATED WORK

Compared to the large body of work on new models for *detection*, prior work on unsupervised OD model *selection* is quite slim. Earlier work proposed intrinsic measures for unsupervised model evaluation, based on input data and output outlier scores Goix (2016); Marques et al. (2015), or based on consensus among various models Duan et al. (2020); Lin et al. (2020), as well as properties of the learned weights Martin et al. (2021). As recent meta-analyses have shown, intrinsic measures are quite noisy; only slightly and often no better than random Ma et al. (2023). Moreover, they suffer from exponential compute cost in large HP spaces as they require training numerous candidate models for evaluation. More recent solutions leverage meta-learning by selecting a model for a new

dataset based on how they perform on similar historical datasets Zhao et al. (2021; 2022); Jiang et al. (2023). They are computationally slow in large HP spaces and cannot handle any continuous HPs. Our proposed HYPER also leverages meta-learning, while it is more efficient with hypernetworks, and is more effective by handling continuous HPs with a well-designed performance estimator function. In experiments, we compare all the aforementioned baselines (see Fig. 5 and Table C2).

## 1.2 PRESENT WORK

We introduce HYPER and tackle two key challenges with unsupervised DOD model selection: **(Ch1)** lack of supervision, and **(Ch2)** scalability as tempered by the cost of training numerous candidate models. For (Ch1), we employ meta-learning where the main idea is to train a performance estimator function, $f_{\text{val}}$, which maps the input data, HPs $\boldsymbol{\lambda}$, and output outlier scores corresponding to $\boldsymbol{\lambda}$ onto detection performance on historical tasks. Note that meta-learning builds on past experience, i.e. historical datasets *with* labels for which detection performance of various models can be evaluated.

Having substituted $\mathcal{L}_{\text{val}}$ with meta-trained $f_{\text{val}}$, one can adopt existing supervised HPO solutions Feurer & Hutter (2019) for model selection on a given/new dataset *without* labels. However, most of those are susceptible to the scalability challenge, as they train each candidate model (with varying $\boldsymbol{\lambda}$) independently from scratch. To address (Ch2) and bypass the expensive process of fully training each candidate separately, we leverage *hypernetworks* (HN). This idea is inspired by the self-tuning networks MacKay et al. (2019), which estimate the best-response function that maps HPs $\boldsymbol{\lambda}$ onto optimal weights $\mathbf{W}^*$ through a hypernetwork (HN) parameterized by $\phi$, i.e. $\widehat{\mathbf{W}}_\phi(\boldsymbol{\lambda}) \approx \mathbf{W}^*$.

A single auxiliary HN model can generate the weights of the main DOD model with varying HPs. In essence, it learns how the model weights should change or *respond to* the changes in HPs (hence the name, best-response). As one of our key contributions, besides the regularization HPs (e.g., dropout rate) that STN MacKay et al. (2019) considered, we propose a novel HN model that can also respond to *architectural* HPs; including depth and width for DOD models with fully-connected layers.

In a nutshell, HYPER jointly optimizes the HPs $\boldsymbol{\lambda}$ and the HN parameters $\phi$ in an alternating fashion. Over iterations, it alternates between (1) **HN-training** that updates $\phi$ to approximate the best-response in a local neighborhood around the current hyperparameters via $\mathcal{L}_{\text{trn}}$, and then the (2) **HPO** step that updates $\boldsymbol{\lambda}$ in a gradient-free fashion by estimating detection performance through $f_{\text{val}}$ of a large set of candidate $\boldsymbol{\lambda}$'s sampled from the same neighborhood, using the corresponding approximate best-response, i.e. the HN-generated weights.

Our HN model offers dramatic speed-ups, by dynamically generating weights for many candidate models with varying HPs, as compared to freely training these candidates separately. Therefore, we also utilize HNs during meta-training, where we replace independently training many models on each historical dataset with a single HN. HN's ability to produce weights for HPs unseen during HN-training also makes it an attractive design choice for continuous-space HPO.

**Summary of contributions.** HYPER addresses the model selection problem for *unsupervised* deep-NN based outlier detection (DOD), applicable to any DOD model, and is *efficient* despite the large continuous HP space including regularization as well as NN architecture HPs. HYPER's notable efficiency is thanks to our proposed hypernetwork (HN) model that generates DOD model parameters (i.e. NN weights) in response to changes in the HPs—in effect, we leverage a single HN acting like many DOD models. Further, it offers unsupervised tuning thanks to a performance estimator function trained via meta-learning on historical tasks, which also benefits from the efficiency of our HN.

We compare HYPER against 8 baselines through extensive experiments on 35 benchmark datasets. HYPER achieves the best performance-runtime trade-off, and selects models with statistically better detection than all baselines (see Fig. 5). Notably, it offers 30% performance improvement over the default HPs, and 7% improvement with $4.2\times$ speed-up over the latest method ELECT Zhao et al. (2022) (see Table C2 in Appx.), thus establishing the new state-of-the-art.

**Accessibility and Reproducibility**. See repo. `https://github.com/inreview23/HYPER`.

## 2 PRELIMINARIES

### 2.1 PROBLEM AND CHALLENGES

The sensitivity of outlier detectors to hyperparameter (HP) choices is well studied Campos et al. (2016a). Deep-NN models are no exception, in fact are even more sensitive as they have many more HPs Ding et al. (2022). In fact, it would not be an overstatement to point to unsupervised outlier

model selection as the primary obstacle to unlocking the ground-breaking potential of deep-NNs for OD. This is exactly the problem we consider in this work.

**Problem 1 (Unsupervised Deep Outlier Model Selection (UDOMS))** Given *a new input dataset (i.e., detection task[1]) $\mathcal{D}_{test} = (\mathbf{X}_{test}, \emptyset)$ without any labels, and a deep-NN based OD model $M$;* Output *model parameters corresponding to a selected hyperparameter configuration $\boldsymbol{\lambda} \in \boldsymbol{\Lambda}$ to employ on $\mathbf{X}_{test}$ to maximize $M$'s detection performance.*

**Desiderata.** Our work tackles two key challenges that arise when tuning OD models with deep NNs: (1) *Validation without supervision*, and (2) *Large HP/model space*.

First, unsupervised OD does not exhibit any labels and therefore model selection via validating detection performance on labeled hold-out data is not possible. While model parameters can be estimated end-to-end through unsupervised training losses, such as reconstruction error or one-class losses, one cannot reliably use the same loss as the validation loss; in fact, low error could easily associate with poor detection since most DOD models use point-wise errors as their outlier scores.

Second, model tuning for the modern OD techniques based on deep-NNs with many HPs is a much larger scale ball-game than that for their shallow counterparts with only 1-2 HPs. This is both due to their ($i$) large number of HPs and also ($ii$) longer training time they typically demand. In other words, the model space that is exponential in the number of HPs and the costly training of individual models necessitate efficient strategies for effective search.

## 2.2 Hypernetworks

We approach the challenge of efficiently searching the HP space with the help of hypernetworks, for which we provide necessary background in this section.

In principle, a hypernetwork (HN) is a (usually small) network generating weights (i.e. parameters) for another larger network (called the main network) Ha et al. (2017). As such, one can think of the HN as a "model compression" tool for training, one that requires fewer learnable parameters. HNs have been used mainly for parameter-efficient training of large models with diverse architectures MacKay et al. (2019),Brock et al. (2018), Zhang et al. (2019),Knyazev et al. (2021) as well as for diverse learning tasks Przewięźlikowski et al. (2022), von Oswald et al. (2020).

Historically, HNs can be seen as the birth-child of the "fast-weights" concept by Schmidhuber (1992), where one network produces *context-dependent weight changes* for another network. The *context*, in our as well as several other work Brock et al. (2018); MacKay et al. (2019), is the *hyperparameters* (HPs).[2] That is, we train a HN model that takes the (encoding of) HPs of the (main) DOD model as input, and produces *HP-dependent* weight changes for the DOD model that we aim to tune. Training a *single* HN that can generate weights for the (main) DOD model for *varying HPs* can effectively bypass the cost of fully-training those candidate models separately. This offers dramatic speed up during model search where one trained (HN) model acts like multiple trained (DOD) models.

## 3 Proposed Framework for UDOMS: HyPer

**Overview.** HyPer consists of two phases (see Fig. 1): (§3.1) offline meta-training over historical datasets, and (§3.2) online model selection for the test dataset. First we train the performance estimator $f_{\text{val}}$ offline, which allows us to predict model performance on the test dataset without relying on any labels. Given a new task online, we alternate between training our HN to efficiently generate model weights for varying HPs around a local HP neighborhood, and refining the best HPs at the current iteration based on $f_{\text{val}}$'s predictions for many locally sampled HPs.

We present the offline and online phases in detail as follows, and later in §4 describe the specifics of our proposed HN, which is employed in both phases for efficient model training.

## 3.1 Meta-Training (Offline on Historical Datasets)

Through meta-learning the goal is to transfer supervision from labeled historical datasets, $\mathcal{D}_{\text{train}} = \{\mathcal{D}_i = (\mathbf{X}_i, \mathbf{y}_i)\}_{i=1}^{n}$, to enable model performance evaluation on a new dataset without labels. To that end, we train $f_{\text{val}}$ to map {`data_embedding`, `model_embedding`, `HP_config`} onto

---

[1]Throughout text, we use outlier detection *task* and *dataset* interchangeably.

[2]We remark that *hyper*networks need not depend in any form to *hyper*parameters, as the naming similarity may (incorrectly) suggest. In fact, earlier work used HNs for model compression rather than model/HP selection.

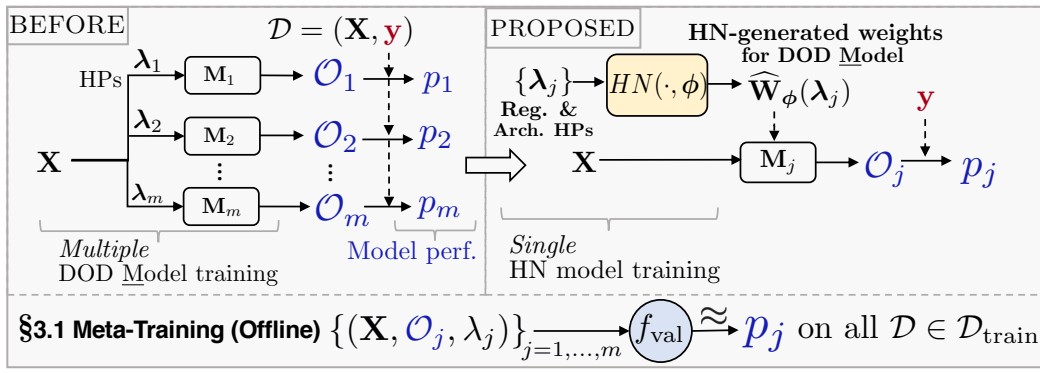

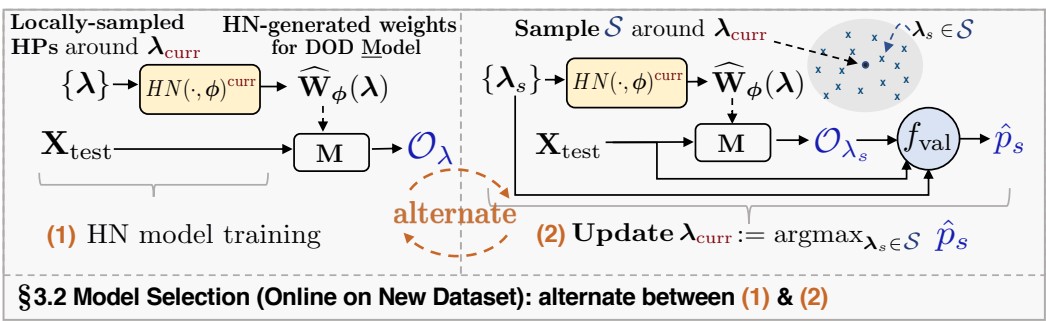

**Figure 1:** HYPER framework illustrated. (top) Offline meta-training of performance estimator $f_{\text{val}}$ (depicted in █) on labeled historical datasets $\mathcal{D}_{\text{train}}$ (§3.1); (bottom) Online model selection on a new unlabeled dataset $\mathbf{X}_{\text{test}}$ (§3.2). We accelerate both meta-training and model selection using hypernetworks (HN) (depicted in █; §4).

the corresponding model performance across $\mathcal{D}_{\text{train}}$. $f_{\text{val}}$ is then employed to predict performance solely from characteristics of (1) the input data, (2) the trained model, and (3) the HP values.

**Data Embedding.** Datasets may have different feature and sample sizes (Appx. §C.1), which makes it challenging to learn dataset embeddings. To address this, we use feature hashing Weinberger et al. (2009), $\psi(\cdot)$, to project a dataset to a $k$-dimensional unified feature space. Then, we train a feature extractor $h(\cdot)$, a fully connected neural network, to map hashed samples to their corresponding outlier labels, i.e. $h : \psi(\mathbf{X}) \mapsto \mathbf{y}$, where $(\mathbf{X}, \mathbf{y})$ denotes a historical dataset with labels. In effect, embeddings by $h(\cdot)$ is expected to capture outlying characteristics of datasets. Finally, we use max-pooling to aggregate sample-wise representations into dataset-wise embeddings by $\text{pool}\{h(\psi(\mathbf{X}))\}$.

**Model Embedding.** To represent a trained DOD model with HP-config. $\boldsymbol{\lambda}$, we train a neural network $g(\cdot)$ that maps its set of output outlier scores onto detection performance, i.e. $g : \mathcal{O}_{\boldsymbol{\lambda}} \mapsto p$. To handle set-size variability of outlier scores (due to different size datasets), we employ DeepSet Zaheer et al. (2017) for $g(\cdot)$, and use the pooling layer's output as the model embedding by $\text{pool}\{g(\mathcal{O}_{\boldsymbol{\lambda}})\}$.

**Training an Effective and Efficient $f_{\text{val}}$ for Validation without Supervision.** Given a DOD algorithm $M$ for UDOMS, let $M_j$ denote the model with HP configuration $\boldsymbol{\lambda}_j$ from the set $\boldsymbol{\lambda}_{\text{meta}} = \{\boldsymbol{\lambda}_1, \dots, \boldsymbol{\lambda}_m\} \in \boldsymbol{\Lambda}$. HYPER uses $\mathcal{D}_{\text{train}}$ to compute (1) *historical outlier scores* $\mathcal{O}_{i,j}$, as output by each $M_j$ on each $\mathcal{D}_i \in \mathcal{D}_{\text{train}}$; and the corresponding (2) *historical performance* $p_{i,j}$, denoting $M_j$'s detection performance (e.g. AUROC) on $\mathcal{D}_i$, calculated based on the scores $\mathcal{O}_{i,j}$ and the labels $\mathbf{y}_i$.

*Regression.* As shown in Fig. 1 (top), the idea of $f_{\text{val}}$ is to learn a mapping (e.g. lightGBM Ke et al. (2017) or any other regressor) from (1) input data embedding, (2) model embedding (based on model output, i.e. outlier scores), and (3) given HP configuration onto the corresponding detection performance across $n$ historical datasets and $m$ models (i.e. configurations). Specifically,

$$f_{\text{val}} : \big\{ \underbrace{\text{pool}\{h(\psi(\mathbf{X}_i))\}}_{\text{data embed.}}, \underbrace{\text{pool}\{g(\mathcal{O}_{i,j})\}}_{\text{model embed.}}, \underbrace{\boldsymbol{\lambda}_j}_{\text{HPs}} \big\} \mapsto p_{i,j} \ , \quad i \in \{1, \dots, n\}, \ j \in \{1, \dots, m\} \ . \quad (2)$$

Further details on $f_{\text{val}}$ are given in Appx. §A. Notably, provided with the functions $\psi(\cdot)$, $h(\cdot)$, $g(\cdot)$, and the trained $f_{\text{val}}$, we can predict a given model's performance on a new task *without* any labels.

*Speed-up.* Obtaining model embeddings requires the outlier scores and hence training the DOD model for each HP configuration, which can be computationally expensive (see Fig. 1 (top, left)). To address this efficiency issue, we train our proposed HN (details presented in §4) only *once per dataset* across $m$ different HP configurations, which generates the weights and outlier scores for all models, significantly speeding up the meta-training phase (see Fig. 1 (top, right)).

---

**Algorithm 1** HYPER: Online Model Selection

---

**Input:** test dataset $\mathcal{D}_{\text{test}} = (\mathbf{X}_{\text{test}}, \emptyset)$ HN parameters $\phi$, HN learning rate $\alpha$, HN loss function $\mathcal{L}_{\text{hn}}(\cdot)$, performance estimator $f_{\text{val}}$, HN (re-)training epochs $T$, validation objective $\mathcal{G}(\cdot)$, patience $\rho$
**Output:** optimized HP configuration $\boldsymbol{\lambda}^*$ for the test dataset

---

1: Initialize $\boldsymbol{\lambda}_{\text{curr}}$ and $\boldsymbol{\sigma}_{\text{curr}}$; set current best HP $\boldsymbol{\lambda}^* := \boldsymbol{\lambda}_{\text{curr}}$; sampled set of HPs $\mathcal{S} := \emptyset$
2: **while** patience criterion $\rho$ is not met **do**
3:    **for** $t = 1, \ldots, T$ **do**
4:       $\boldsymbol{\epsilon} \sim p(\boldsymbol{\epsilon}|\boldsymbol{\sigma}_{\text{curr}})$                 ▶ sample local HP perturbations around current $\boldsymbol{\lambda}_{\text{curr}}$
5:       $\phi \leftarrow \alpha \frac{\partial}{\partial \phi} \mathcal{L}_{\text{hn}}(\boldsymbol{\lambda}_{\text{curr}} + \boldsymbol{\epsilon}, \widehat{\mathbf{W}}_\phi(\boldsymbol{\lambda}_{\text{curr}} + \boldsymbol{\epsilon}))$     ▶ train the HN with the sampled local HPs
6:       $\mathcal{S} := \mathcal{S} \cup (\boldsymbol{\lambda}_{\text{curr}} + \boldsymbol{\epsilon})$                      ▶ save locally sampled HPs
7:    **end for**
8:    $\boldsymbol{\lambda}_{\text{curr}} \leftarrow \operatorname{argmax}_{\boldsymbol{\lambda} \in \boldsymbol{\Lambda}} \mathcal{G}(\boldsymbol{\lambda}, \boldsymbol{\sigma}_{\text{curr}}, \widehat{\mathbf{W}}_\phi(\boldsymbol{\lambda} + \boldsymbol{\epsilon}))$            ▶ update $\boldsymbol{\lambda}_{\text{curr}}$ by Eq. (4)
9:    $\boldsymbol{\sigma}_{\text{curr}} \leftarrow \operatorname{argmax}_{\boldsymbol{\sigma}} \mathcal{G}(\boldsymbol{\lambda}_{\text{curr}}, \boldsymbol{\sigma}, \widehat{\mathbf{W}}_\phi(\boldsymbol{\lambda} + \boldsymbol{\epsilon}))$            ▶ update $\boldsymbol{\sigma}_{\text{curr}}$ by Eq. (4)
10: **end while**
11: Output the best HP $\boldsymbol{\lambda}^* \approx \operatorname{argmax}_{\boldsymbol{\lambda} \in \mathcal{S}} f_{\text{val}}(\mathbf{X}_{\text{test}}, \widehat{\mathbf{W}}_\phi(\boldsymbol{\lambda}), \boldsymbol{\lambda})$          ▶ Eq. (5)

---

## 3.2 MODEL SELECTION (ONLINE ON NEW DATASET)

**Model selection via performance estimator.** Given our meta-trained $f_{\text{val}}$, we can train DOD models with randomly sampled HPs on the test dataset to obtain outlier scores, and then select the one with the highest predicted performance by $f_{\text{val}}$, that is, $\operatorname{argmax}_{\boldsymbol{\lambda} \in \boldsymbol{\Lambda}} f_{\text{val}}(\mathbf{X}_{\text{test}}, \mathcal{O}_{\text{test}, \boldsymbol{\lambda}}, \boldsymbol{\lambda})$ .

Training OD models from scratch for each HP can be computationally expensive. To speed this up, we propose to build a HN to generate model weights and subsequently the outlier scores for randomly sampled HPs. Parameterized by $\phi$, the HN maps a given HP configuration $\boldsymbol{\lambda}_j$ to the weights $\widehat{\mathbf{W}}_\phi(\boldsymbol{\lambda}_j) := HN(\boldsymbol{\lambda}_j; \phi)$, which are effectively the predicted parameters of the DOD model under HP configuration $\boldsymbol{\lambda}_j$. (See details in §4.) As the DOD model weights also dictate the output outlier scores $\mathcal{O}_{\boldsymbol{\lambda}}$, we abuse notation and use them interchangeably as input to $f_{\text{val}}$ in this section.

**Training local HN iteratively and adaptively.** We propose to iteratively train our HN over *locally* selected HPs, since training a "global HN" to predict weights across the entire $\boldsymbol{\Lambda}$ and over unseen $\boldsymbol{\lambda}$ is a challenging task especially for large model spaces MacKay et al. (2019), impacting the quality of model selection. We design HYPER to jointly optimize the HPs $\boldsymbol{\lambda}$ and the (local) HN parameters $\phi$ in an alternating fashion; as shown in Fig. 1 (bottom) and Algo. 1. It alternates between:

1. **HN-training** that updates HN parameters $\phi$ to approximate the best-response in a local neighborhood around the current hyperparameters $\boldsymbol{\lambda}_{\text{curr}}$ via $\mathcal{L}_{\text{hn}}$, and

2. **HPOpt** that updates $\boldsymbol{\lambda}_{\text{curr}}$ in a gradient-free fashion by estimating detection performance through $f_{\text{val}}$ of a large set $\mathcal{S}$ of candidate $\boldsymbol{\lambda}$'s sampled from the same neighborhood, using the corresponding approximate best-response, i.e. the HN-generated weights, $\widehat{\mathbf{W}}_\phi(\boldsymbol{\lambda})$.

To dynamically control the sampling range around $\boldsymbol{\lambda}_{\text{curr}}$, we use a factorized Gaussian with standard deviation $\boldsymbol{\sigma}$ to generate local HP perturbations $p(\boldsymbol{\epsilon}|\boldsymbol{\sigma})$. $\boldsymbol{\sigma}_{\text{curr}}$ is used during HN-training for sampling local HPs and gets updated during HPOpt at each iteration.

**Updating $\boldsymbol{\lambda}_{\text{curr}}$ and $\boldsymbol{\sigma}_{\text{curr}}$.** HYPER iteratively explores promising HPs and the corresponding sampling range. To update $\boldsymbol{\lambda}_{\text{curr}}$ and $\boldsymbol{\sigma}_{\text{curr}}$, we maximize the following objective.

$$\underbrace{\mathbb{E}_{\boldsymbol{\epsilon} \sim p(\boldsymbol{\epsilon}|\boldsymbol{\sigma})}[f_{\text{val}}(\mathbf{X}_{\text{test}}, \widehat{\mathbf{W}}_\phi(\boldsymbol{\lambda} + \boldsymbol{\epsilon}), \boldsymbol{\lambda} + \boldsymbol{\epsilon})]}_{\text{update } \boldsymbol{\lambda}_{\text{curr}} \text{ to a better model/HPs w/ high expectation}} \quad + \quad \underbrace{\tau \, \mathbb{H}(p(\boldsymbol{\epsilon}|\boldsymbol{\sigma}))}_{\text{sampling range around } \boldsymbol{\lambda}_{\text{curr}}} \tag{3}$$

The objective consists of two terms. The first term emphasizes selecting the next model/HP configuration with high expected performance, aiming to improve the overall model performance. The second term measures the uncertainty of the sampling factor, quantified with Shannon's entropy $\mathbb{H}$. A higher entropy value indicates a less localized sampling, allowing for more exploration. The objective is to find an HP configuration that can achieve high expected performance, within a reasonably local region to contain a good model, that is also local enough for the HN to be able to effectively learn the best-response. If the sampling factor $\boldsymbol{\sigma}$ is too small, it limits the exploration of the next HP configuration and training of the HN, potentially missing out on better-performing options. Conversely, if $\boldsymbol{\sigma}$ is too large, it may lead to inaccuracies in the HN's generated weights, compromising the accuracy of the first term. The balance factor $\tau$ controls the trade-off between the two terms.

We approximate the expectation term in Eq. (3) by the empirical mean of predicted performances through $V$ number of sampled perturbations around $\boldsymbol{\lambda}$. Then, we define our validation objective $\mathcal{G}$ as

$$\mathcal{G}(\boldsymbol{\lambda}, \boldsymbol{\sigma}, \widehat{\mathbf{W}}_{\boldsymbol{\phi}}) = \frac{1}{V} \sum_{i=1}^{V} f_{\text{val}}(\mathbf{X}_{\text{test}}, \widehat{\mathbf{W}}_{\boldsymbol{\phi}}(\boldsymbol{\lambda} + \boldsymbol{\epsilon}_i), \boldsymbol{\lambda} + \boldsymbol{\epsilon}_i) + \tau \mathbb{H}(p(\boldsymbol{\epsilon}|\boldsymbol{\sigma})) \ . \tag{4}$$

In each iteration of the HP configuration update, we first fix $\boldsymbol{\sigma}_{\text{curr}}$ and find the HP configuration with the highest value of Eq. (4). Specifically, we sample $V_{\lambda}$ local configurations around $\boldsymbol{\lambda}_{\text{curr}}$, i.e., $\boldsymbol{\lambda}_{\text{curr}} + \boldsymbol{\epsilon}_i|\boldsymbol{\sigma}_{\text{curr}}$ for $i \in 1, \dots, V_{\lambda}$. After $\boldsymbol{\lambda}_{\text{curr}}$ is updated, we fix it and update the sampling factor $\boldsymbol{\sigma}_{\text{curr}}$ by Eq. (4) based on $V_{\sigma}$ samples of $\boldsymbol{\sigma}$. To ensure encountering a good HP configuration, we set $V_{\lambda}$ and $V_{\sigma}$ to be a large number, e.g. 500. (see specific settings in Appx. §C.2.)

**Selecting the Best Model/HP $\boldsymbol{\lambda}^*$.** We employ $f_{\text{val}}$ to choose the best HP $\boldsymbol{\lambda}^*$ from all the locally sampled HPs $\mathcal{S}$ during the HN training. Note that HYPER directly uses the HN-generated weights $\widehat{\mathbf{W}}_{\boldsymbol{\phi}}(\boldsymbol{\lambda})$ for fast computation, without the need to build any model for evaluating by $f_{\text{val}}$. That is,

$$\boldsymbol{\lambda}^* \approx \underset{\boldsymbol{\lambda} \in \mathcal{S}}{\operatorname{argmax}} \ f_{\text{val}}(\mathbf{X}_{\text{test}}, \widehat{\mathbf{W}}_{\boldsymbol{\phi}}(\boldsymbol{\lambda}), \boldsymbol{\lambda}) \ . \tag{5}$$

**Initialization and Convergence.** We initialize $\boldsymbol{\lambda}_{\text{curr}}$ and $\boldsymbol{\sigma}_{\text{curr}}$ with the globally best values across historical datasets. We consider HYPER as converged if the highest predicted performance by $f_{\text{val}}$ does not change in $\rho$ consecutive iterations. A larger $\rho$, referred as "patience", requires more iterations to converge yet likely yields better results. Note that $\rho$ can be decided by cross-validation on historical datasets during meta-training. See Appx. §C.3 for analysis of initialization and patience.

## 4 PROPOSED HYPERNETWORK FOR SPEED UP – TRAIN ONE, GET MANY

To tackle the challenge of model-building efficiency, we design hypernetworks (HN) that can efficiently train OD models with different hyperparameter configurations. A hypernetwork (HN) is a network generating weights (i.e. parameters) for another network (in our case, the DOD model) Ha et al. (2017). Our input to HN, $\boldsymbol{\lambda} \in \boldsymbol{\Lambda}$, breaks down into two components as $\boldsymbol{\lambda} = [\boldsymbol{\lambda}_{reg}, \boldsymbol{\lambda}_{arch}]$, corresponding to regularization HPs (e.g. dropout, weight decay) and architectural HPs.

Our HN resolves three challenges: **(Ch.I)** its fixed-size output $\widehat{\mathbf{W}}_{\boldsymbol{\phi}}$ must be able to adjust to different architectural shapes, **(Ch.II)** $\widehat{\mathbf{W}}_{\boldsymbol{\phi}}(\boldsymbol{\lambda})$ should output sufficiently diverse weights in response to varying $\boldsymbol{\lambda}$ inputs, and **(Ch.III)** training HN should be more efficient than training individual DOD models.

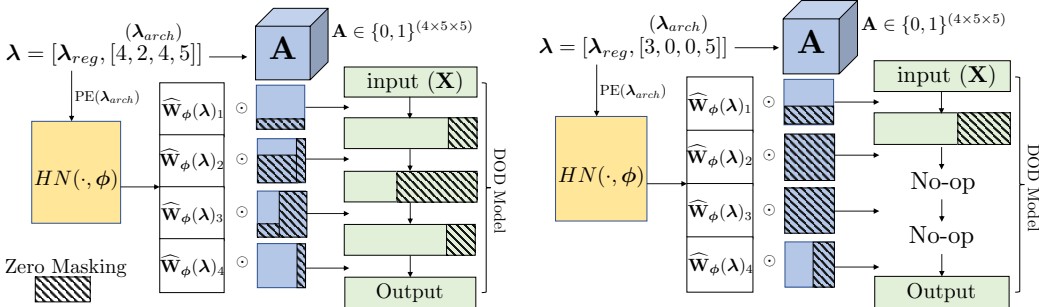

**Figure 2:** Illustration of the proposed HN. Left: HN generates weights for a 4-layer AE, with layer widths equal to $[4, 2, 4, 5]$. Weights $\widehat{\mathbf{W}}_{\boldsymbol{\phi}}$ is fed into the DOD model, while hidden layers' dimensions are shrunk by the masking $\mathbf{A}$. Right: HN generates weights for a 2-layer AE, with layer widths equal to $[3, 5]$. $\boldsymbol{\lambda}_{arch}$ is padded as $[3, 0, 0, 5]$, and the architecture masking at the second and third layer are set to all zeros. When $\widehat{\mathbf{W}}_{\boldsymbol{\phi}}$ is fed into the DOD model, zero masking enables the No-op, in effect shrinking the DOD model from 4 layers to 2 layers.

**Architecture Masking for (Ch.I)** To allow HN output to adapt to various architectures, we let $\widehat{\mathbf{W}}_{\boldsymbol{\phi}}$'s size be equal to size of the largest architecture in model space $\boldsymbol{\Lambda}$. Then for each $\boldsymbol{\lambda}_{arch}$, we build a corresponding architecture masking $\mathbf{A}$ and feed the $\mathbf{A}$-masked version of $\widehat{\mathbf{W}}_{\boldsymbol{\phi}}$ to the DOD model. In other words, our $\widehat{\mathbf{W}}_{\boldsymbol{\phi}}$ handles all smaller architectures by properly padding zeros on the $\widehat{\mathbf{W}}_{\boldsymbol{\phi}}$.

Taking DOD models built upon MLPs as an example (see Fig. 2), we make HN output $\widehat{\mathbf{W}}_{\boldsymbol{\phi}} \in \mathbb{R}^{(D \times W \times W)}$, where $D$ and $W$ denote the maximum depth and maximum layer width from $\boldsymbol{\Lambda}$. Assume $\boldsymbol{\lambda}_{arch}$ contains the abstraction of a smaller architecture; e.g., $L$ layers with corresponding

width values $\{W_1, W_2 \ldots, W_L\}$ all less than or equal to $W$. Then $\boldsymbol{\lambda}_{arch} \in \mathbb{N}^D$ is given as

$$\boldsymbol{\lambda}_{arch} = [W_1, W_2, \ldots, W_{\lfloor L/2 \rfloor}, \underbrace{0, \ldots, 0}_{(D-L) \text{ zeros}}, W_{\lfloor L/2 \rfloor + 1}, \ldots, W_{(L-1)}, W_L].$$

The architecture masking $\mathbf{A} \in \{0,1\}^{(D \times W \times W)}$ is constructed as the following:

$$\begin{cases} \mathbf{A}_{[l, 0:\boldsymbol{\lambda}_{arch}[0], :]} = 1 & \text{, if } l = 0 \\ \mathbf{A}_{[l, 0:\boldsymbol{\lambda}_{arch}[l], 0:\boldsymbol{\lambda}_{arch}[l-z]]} = 1 & \text{, otherwise} \end{cases} \tag{6}$$

where $\boldsymbol{\lambda}_{arch}[l-z]$ is the last non-zero entry in $\boldsymbol{\lambda}_{arch}[0:l]$ (e.g., for $\boldsymbol{\lambda}_{arch} = [5, 3, 0, 0, 3]$ and $l = 4$, the last nonzero entry is $\boldsymbol{\lambda}_{arch}[1]$ where $z = 3$). Then, $l$'th layer weights are multiplied by masking as $\mathbf{A}_{[l,:,:]} \odot \widehat{\mathbf{W}}_{\phi,l}$, where non-zero entries are of shrunk dimensions. If $\mathbf{A}_{[l,:,:]}$ contains only zeros, layer weights become all zeros, representing a "No-op" (and DOD model ignores this layer).

**Other Architectures.** We find that this masking works well with linear autoencoders with a "hourglass" structure, in which case the maximum width $W$ is the input dimension. For networks built with convolutions, on the other hand, the architecture masking becomes $\mathbf{A} \in \{0,1\}^{D \times M_{ch} \times M_{ch} \times M_k \times M_k}$, where $D, M_{ch}, M_k$ represent maximum number of layers, channels, and kernel size specified in $\boldsymbol{\Lambda}$, respectively. Here we abbreviate the masking procedure: Channels, similar to the previously discussed widths, are padded by masking out the second dimension of the $\mathbf{A}$ tensor. When we need a smaller kernel size $k \leq M_k$ at layer $l$, the corresponding $\mathbf{A}_{[l,:,:,:]}$ pads zeros around the size $M_{ch} \times k \times k$ center. The masked weights $\mathbf{A}_{[l,:,:,:]} \odot \widehat{\mathbf{W}}_{\phi,l}$ are equivalent to obtaining smaller-size kernel weights, as shown in Wang et al. (2020). Further details of masking and constructing the $\boldsymbol{\lambda}_{arch}$ input for other architectures can be found in Appx. §B.

**Diverse Weight Generation for (Ch.II)** While HN is a universal function approximator in theory, it may not generalize well to offer good approximations for many unseen architectures ], especially given that the number of $\boldsymbol{\lambda}$'s during training is limited. When there is only little variation between two inputs $\boldsymbol{\lambda}_j$ and $\boldsymbol{\lambda}_{j'}$, the HN provides more similar weights $\widehat{\mathbf{W}}_\phi(\boldsymbol{\lambda}_j)$ and $\widehat{\mathbf{W}}_\phi(\boldsymbol{\lambda}_{j'})$, since the weights are generated from the same HN where implicit weight sharing occurs.

We employ two ideas toward enabling the HN to generate more expressive weights in response to changes in $\boldsymbol{\lambda}_{arch}$. First is to inject more variation within its input space where, instead of directly feeding in $\boldsymbol{\lambda}_{arch}$, we input the positional encoding of each element in $\boldsymbol{\lambda}_{arch}$. Positional encoding Vaswani et al. (2017) transforms each scalar element into a vector embedding, which encodes more granular information, especially when $\boldsymbol{\lambda}_{arch}$ contains zeros representing a shallower sub-architecture. Second idea is to employ a scheduled training strategy of the HN as it produces weights for both shallow and deep architectures. During HN training, we train with $\boldsymbol{\lambda}$ associated with deeper architectures first, and later $\boldsymbol{\lambda}$ for shallower architectures are trained jointly with deeper architectures. Our scheduled training alleviates the

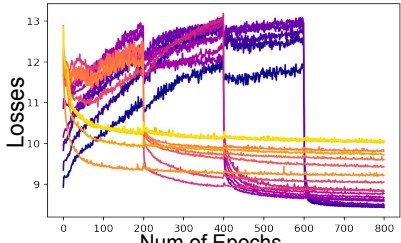

**Figure 3:** Loss of individual models during scheduled training. Lighter colors depict loss curves of deeper architectures, which enter training early.

problem of imbalanced weight sharing, where weights associated with shallower layers are updated more frequently as those are used by more number of architectures. Fig. 3 (best in color) illustrates how the training losses change for individual architectures during the HN's scheduled training.

**Batchwise Training for (Ch.III)** Like other NNs, HN allows for several inputs $\{\boldsymbol{\lambda}_j\}_{j=1}^m$ synchronously and outputs $\{\widehat{\mathbf{W}}_\phi(\boldsymbol{\lambda}_j)\}_{j=1}^m$. To speed up training, we batch the input at each forward step with a set of different architectures $\mathcal{M}$ to obtain $\{\boldsymbol{\lambda}_{arch,j}\}_{j \in \mathcal{M}}$, which pair with a sampled regularization HP configuration, $\boldsymbol{\lambda}_{reg,s}$. Given dataset $\mathcal{D}$, the HN loss for one pass is calculated as

$$\mathcal{L}_{hn} = \sum_{\mathbf{x} \in \mathcal{D}} \sum_{j \in \mathcal{M}} \mathcal{L}_{trn}\left(\widehat{\mathbf{W}}_\phi([\boldsymbol{\lambda}_{arch,j}, \boldsymbol{\lambda}_{reg,s}]), \mathbf{x}\right). \tag{7}$$

In summary, our HN mimics fast DOD model building across different HP configurations. This offers two advantages: ($i$) training many different HPs jointly in meta-training and ($ii$) fast DOD model parameter generation during online model search. Notably, our HN can tune a wider range of HPs including model architecture, and as shown in §5.2, provides superior results to only tuning $\boldsymbol{\lambda}_{reg}$.

## 5 EXPERIMENTS

**Benchmark Data.** We show HYPER's effectiveness and efficiency with fully connected AutoEncoder (AE) for DOD on tabular data, using a testbed consisting of 35 benchmark datasets from two different public OD repositories; ODDS Rayana (2016) and DAMI Campos et al. (2016b)). See Appx. C.1 for detailed descriptions of the datasets in both repositories.

**Baselines.** We include 8 baselines for comparison ranging from simple to state-of-the-art (SOTA); Appx. Table C2 provides a conceptual comparison of the baselines. Briefly, they are organized as

- (*i*) *No model selection*: **(1) Default** adopts the default HPs used in the popular OD library PyOD Zhao et al. (2019), **(2) Random** picks an HP randomly (we report expected performance);
- (*ii*) *Model selection without meta-learning*: **(3) MC** Ma et al. (2023) leverages consensus; and
- (*iii*) *Model selection by meta-learning*: **(4) Global Best (GB)** selects the best performing model on the historical datasets on average, and SOTA baselines include **(5) ISAC** Kadioglu et al. (2010), **(6) ARGOSMART (AS)** Nikolic et al. (2013), **(7) MetaOD** Zhao et al. (2021), and finally the latest SOTA **(8) ELECT** Zhao et al. (2022).

Baselines (1), (2), and (4)-(7) are zero-shot that do not require any candidate model building during model selection. More detailed descriptions of the baselines are given in Appx. §C.2.

**Evaluation.** We use 5-fold cross-validation to split the train/test datasets; that is, each time we use 28 datasets as the historical datasets to select models on the remaining 7 datasets. We use the area under the ROC curve to measure detection performance, while it can be substituted with any other measure. As the raw ROC performances are not comparable across datasets with varying difficulty, we report the normalized ROC *Rank* of an HP/model, ranging from 0 (the best) to 1 (the worst)—i.e., the lower the better. We use the paired Wilcoxon signed rank test Groggel (2000) across all datasets in the testbed to compare two methods. Full performance results on all 35 datasets are in Appx. §C.4.

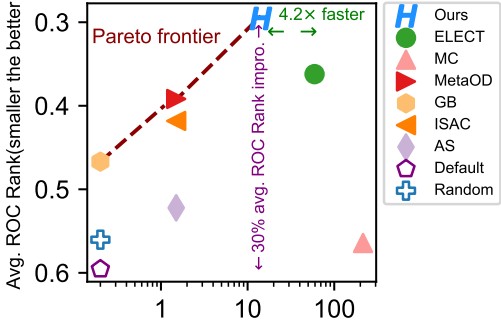

**Figure 4:** Avg. running time (log-scale) vs. avg. model ROC Rank. Meta-learning methods are depicted with solid markers. Pareto frontier (red dashed line) shows the best methods under different time budgets. HYPER outperforms all with reasonable computational demand.

### 5.1 EXPERIMENT RESULTS

Fig. 4 shows that HYPER **outperforms all baselines with regard to the average ROC Rank on the 35 dataset testbed**. It also strikes a good balance between computation and performance.

In addition, Fig. 5 provides the full performance distribution across all datasets and shows that HYPER is statistically better than all baselines, including SOTA meta-learning based ELECT and MetaOD. Among the zero-shot baselines, Default and Random perform significantly poorly while the meta-learning based GB leads to comparably higher performance. Replicating earlier findings Ma et al. (2023), the internal consensus-based MC, while computationally demanding, is no better than Random.

**HN-powered efficiency enables HYPER to search more broadly**. Fig. 4 shows that HYPER offers significant runtime gains over the SOTA method ELECT, with an average of-

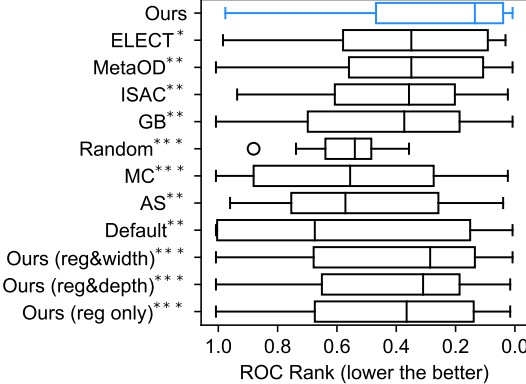

**Figure 5:** Distribution of ROC Rank across datasets. HYPER achieves the best performance among all. Bottom three bars depict HYPER's variants that do not fully tune architectural HPs (for ablation). Paired significance test results are depicted as *significance at 0.1, **at 0.01, ***at 0.001. See $p$-values in Appx. Table C3.

fline training speed-up of $5.8\times$ and a model selection speed-up of $4.2\times$. Unlike ELECT, which requires building OD models from scratch during both offline and online phases, HYPER leverages the HN-generated weights to avoid costly model training for each candidate HP.

Moreover, HYPER can search over a broader range of the HP space thanks to the lower model training cost by HN. This contributes to its effectiveness, yielding 7% improvement in avg. ROC Rank over ELECT and 10% improvement over the latest meta-learning baseline MetaOD (see Table C2 Appx.).

**Meta-learning methods achieve the best performance at different budgets**. Fig. 4 shows that the best performers at different time budgets are global best (GB), MetaOD, and HYPER, which are all on the Pareto frontier. In contrast, simple no-model-selection approaches, i.e., Random and Default, are among the lowest performing methods. Based on Table C2, HYPER achieves respectively 20% and 30% improvements in avg. ROC Rank over Random picking and Default HPs in PyOD Zhao et al. (2019), a widely used open-source OD library. Although meta-learning entails additional (offline) training time, it is to amortize across multiple future model selection tasks in the long run.

## 5.2 ABLATION STUDIES (SEE OTHERS IN APPX. §C.3)

**Benefit of Tuning Architectural HPs via HN**. HYPER tackles the challenging task of accommodating architectural HPs. Through ablations, we study the benefit of our novel HN design, as presented in §4, which can generate DOD model weights in response to changes in architectural HPs. Bottom three bars of Fig. 5 show the performances of three HYPER variants across datasets. The proposed HYPER (with median ROC Rank = 0.1349) outperforms all these variants significantly (with $p<0.001$), namely, *only tuning regularization and width* (median ROC Rank = 0.2857), *only tuning regularization and depth* (median ROC Rank = 0.3095), and *only tuning regularization* (median ROC Rank = 0.3650). By extending its search for both neural network depth *and* width, HYPER explores a larger model space that helps find better-performing model configurations.

**HP Schedules over Iterations**. In Fig. 6, we more closely analyze how the HP values change over HYPER iterations on `spamspace`, where we compare between (top) only tuning the reg. HPs, namely the dropout and weight decay rates, while fixing model depth and width (i.e., shrinkage rate) and (bottom) tuning all HPs including both reg. and architectural HPs. Bottom figures show that depth remains fixed at 4, shrinkage rate increases from 1 to 2.25 (i.e., width gets reduced), while dropout and weight decay reduce to 0.2, and 0.05 respectively; in other words, overall model capac-

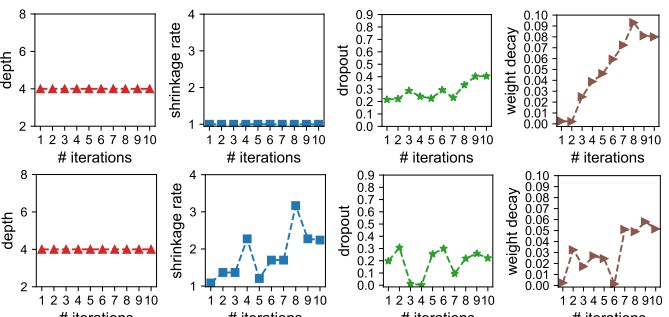

**Figure 6:** Trace of HP schedules over HYPER iterations on `spamspace`: (top) tuning only regularization HPs; (bottom) tuning both regularization and architectural HPs (proposed). When architecture is fixed, reg. HPs incur more changes in magnitude and reach larger values to adjust model complexity. HYPER tunes complexity more flexibly by also accommodating architectural HPs.

ity is reduced relative to initialization. In contrast, top figures show that when model depth and width are fixed, regularization HPs compensate more to regulate the model capacity, with a larger dropout rate at 0.4 and larger weight decay at 0.08, achieving ROC rank of only 0.3227 (top) in contrast to proposed HYPER's 0.0555 (bottom). This comparison showcases the merit of HYPER which adjusts model complexity more flexibly by being able to accommodate a larger model space.

## 6 CONCLUSION

We introduced HYPER, a new framework for unsupervised deep outlier model selection. HYPER tackles two key challenges in this setting: validation in the absence of labels and efficient search of the large hyperparameter (HP)/model space. To that end, it uses meta-learning to train a performance estimator $f_{val}$ on historical datasets to effectively predict model performance on a new task without labels. To speed up search, it utilizes a novel hypernetwork (HN) design that generates weights for the detection model with varying HPs including its architecture, and achieves significant efficiency gains over individually training the candidate models. Experiments on a testbed of 35 benchmark datasets showed that HYPER significantly outperforms all 8 baselines, establishing the new state-of-the-art.

We also discuss limitations and possible extensions of HYPER: Future work can aim to design a better $f_{val}$ estimator, e.g. by improving dataset and model representations; explore other HP optimization strategies for search, which was not our main focus; and develop other HN designs with lower memory requirement as our focus has not been a small HN that also provides model compression.

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
