APPENDIX

## A   META-TRAINING DETAILS

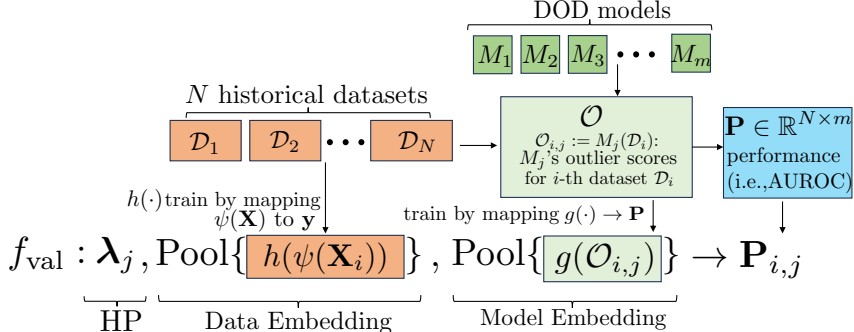

**Figure A1:** Illustration of the $f_{\text{val}}$. We use all historical datasets (i.e., $\mathbf{X}$ and ground-truth labels $\mathbf{y}$) to train the data embeddings and historical embeddings, during which we obtain performance $\mathbf{P}$ on the DOD models' outlier scores $\mathcal{O}$. During the offline phase, We train $h(\cdot)$, $g(\cdot)$, and $f_{\text{val}}$. During the online model selection time, $f_{\text{val}}$ gives the performance estimation of a DOD model (with a specific HP $\boldsymbol{\lambda}$), on the embedding of the test dataset.

We provide details of training $f_{\text{val}}$ in this section. Fig. A1 illustrate the training of $f_{\text{val}}$ in Eq. (2).

**Data Embeddings.** . We employ feature hashing Weinberger et al. (2009) to address the issue of that datasets may have different feature and sample sizes. Specifically, a hashing function $\psi(\cdot)$, is employed to project each dataset to a $k$-dimensional unified feature space, regardless of the number of features in the original data. To ensure sufficient expressiveness, the projection dimension should not be too small (e.g., $k = 256$ in our experiments).

Subsequently, we train a cross-dataset feature extractor $h(\cdot)$, a fully connected neural network, to map hashed samples to their corresponding outlier labels, i.e. $h : \psi(\mathbf{X}_i) \mapsto \mathbf{y}_i$ for the $i$-th dataset. Finally, we use max-pooling to aggregate sample-wise representations into dataset-wise embeddings, denoted by $\text{pool}\{h(\psi(\mathbf{X}_i))\}$. The overall goal of $h(\cdot)$ is to capture outlying characteristics of datasets, as demonstrated in the left middle of Fig. A1.

**Model Embeddings** . In addition to data embeddings, we have also designed model embeddings to capture the impact of model changes on the detection performance. To achieve this, existing work uses internal performance measures (IPMs) as model embeddings Zhao et al. (2022). IPMs are a set of unsupervised performance evaluation metrics for OD that solely rely on model outputs and/or input features. They serve as weak proxies for model performance (more details can be found in Ma et al. (2023)). However, the design of IPMs is typically handcrafted, and their computation at runtime can be computationally expensive.

To represent a trained DOD model more systematically, we utilize a neural network denoted as $g(\cdot)$, which maps the output outlier scores of the model to the corresponding detection performance, i.e. $g : \mathcal{O}_{i,j} \mapsto \mathbf{P}_{i,j}$. To handle the variability in the size of outlier scores, which can arise from differences in sample sizes across datasets, we employ the DeepSet architecture Zaheer et al. (2017) for $g(\cdot)$. The DeepSet architecture is designed to leverage the inherent permutation invariance of sets, meaning that the order of elements in a set does not affect its overall meaning. Similarly, the order of outlier scores does not impact the overall detection performance. In our approach, we use the output of the pooling layer in the DeepSet architecture, denoted as $\text{pool}(g(\mathcal{O}_{i,j}))$, as the model embedding. This pooling layer output effectively captures the information from the outlier scores and produces a fixed-size representation of the model's performance, as shown in the right middle of Fig. A1.

Obtaining outlier scores $\mathcal{O}_{i,j}$ requires training the DOD model, which can be computationally expensive. To speed up this process for meta-training, we use HN-generated weights other than training individual DOD models from scratch, to obtain $\widehat{\mathcal{O}}_{i,j} := M_j(\mathcal{D}_i; \widehat{\mathbf{W}}_{\boldsymbol{\phi}}^{(i)}(\boldsymbol{\lambda}_j))$, where $\widehat{\mathbf{W}}_{\boldsymbol{\phi}}^{(i)}(\boldsymbol{\lambda}_j)$ denotes model $M_j$'s weights as generated by our HN trained on $\mathcal{D}_i$, and $\boldsymbol{\lambda}_j$ are over existing $\boldsymbol{\lambda}_{\mathbf{meta}}$.

**Training $f_{\mathbf{val}}$.** The goal of $f_{\text{val}}$ is to map the aforementioned components, e.g., HPs, data embeddings, and model embeddings, onto the corresponding model performance across $N$ historical datasets and

$m$ models with varying HP configurations. The choice of $f_{\text{val}}$ can be flexible: we use lightGBM Ke et al. (2017) in this work; although one may use any regressor.

We decide the hyperparameters associated with $\psi(\cdot)$, $h(\cdot)$, $g(\cdot)$, and $f_{\text{val}}$ by the cross-validation of the historical datasets. The goal is to optimize the performance of $f_{\text{val}}$ on the historical datasets. We provide additional meta-training details in §C.2.

## B    HYPERNETWORK DETAILS

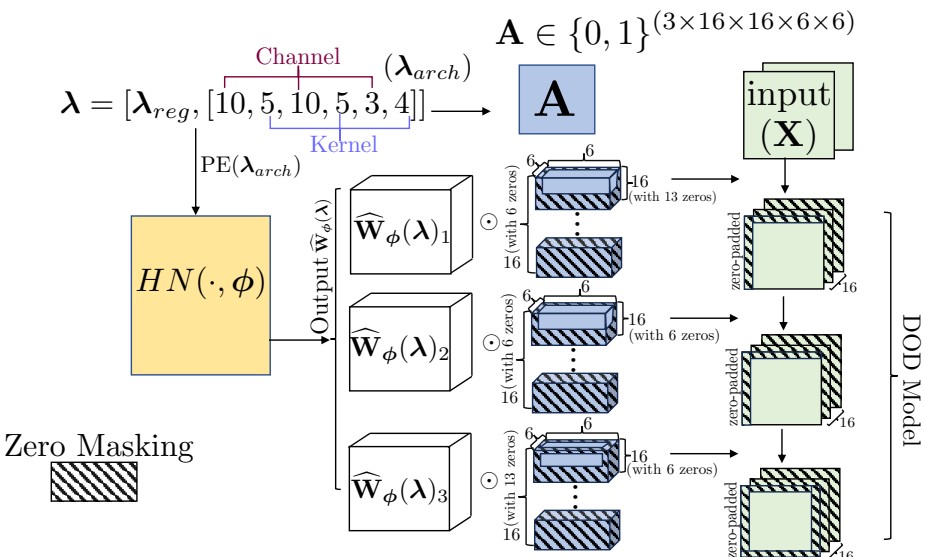

**Figure B2:** Illustration of the proposed HN. HN generates weights for a 3-layer convolutional networks , with channels equal to $[10, 10, 3]$, and kernels equal to $[5, 5, 4]$. The HN weights $\widehat{\mathbf{W}}_\phi$ is of size $3 \times 16 \times 16 \times 6 \times 6$, and similarly we construct the same-size architecture masking $\mathbf{A}$. At the first layer, we need to pad $\mathbf{A}$ for 1 zero, among the third and fourth dimension (we pad starting from the left and from the top). This will enable us to extend $\widehat{\mathbf{W}}_\phi$ to a convolutional operation of kernel size 5, from fixed kernel size 6. To match the padding operation, we also pad the input $\mathbf{X}$ along the first and second dimension, with 1. The rest layers follow similairly.

**Architecture Masking for Convolution Operations.** Since more complex data such as images and videos are used as mediums to find anomalies, many DOD models have taken convolutional networks as the backbone structure. Therefore, how to tune convolution operation within the DOD model has been an emergent problem. Here we design the input of the HN $\lambda_{arch}$ and the architecture $\mathbf{A}$ in a format that could alter output of the HN output $\widehat{\mathbf{W}}_\phi$ to adapt to various architectures.

For convolutional networks, despite that we are able to tune depths and channels, we can also include kernel sizes and dilation rate by properly padding $\widehat{\mathbf{W}}_\phi$ with zeros Wang et al. (2020). For our demonstration purposes, we only inlude the kernel size as the additional turnable variable. We make HN output $\widehat{\mathbf{W}}_\phi \in \mathbb{R}^{(D \times M_{ch} \times M_{ch} \times M_k \times M_k)}$, where $D$, $M_{ch}$, $M_k$ represent maximum number of layers, channels, and kernel size specified in $\mathbf{\Lambda}$, respectively.

Assume $\lambda_{arch}$ contains the abstraction of a smaller architectures, e.g. $L$ layers with corresponding channel values $\{M_{c1}, M_{c2}, ..., M_{cL}\}$ all less than or equal to $M_{ch}$, and $\{K_1, K_2, ..., K_L\}$ are less than or equal to $M_k$. Then, the $\lambda_{arch} \in \mathbb{N}^{2D}$ is given as:

$$\lambda_{arch} = [M_{c1}, K_1, M_{c2}, K_2, \ldots, M_{c\lfloor L/2 \rfloor}, K_{\lfloor L/2 \rfloor}, \underbrace{0, \ldots, 0}_{2(D-L) \text{ zeros}}, M_{c\lfloor L/2 \rfloor+1}, K_{\lfloor L/2 \rfloor+1}, \ldots, M_{cL}, K_L].$$

The architecture masking $\mathbf{A} \in \{0, 1\}^{(D \times M_{ch} \times M_{ch} \times M_k \times M_k)}$ is constructed as the following:

$$\begin{cases} \mathbf{A}_{[l, 0:\lambda_{arch}[2 \times l], :, \lfloor \frac{M_{ch}}{2} \rfloor - \lfloor \frac{\lambda_{arch}[2 \times l+1]}{2} \rfloor : \lfloor \frac{M_{ch}}{2} \rfloor + \lfloor \frac{\lambda_{arch}[2 \times l+1]}{2} \rfloor]} = 1 & , \text{if } l = 0 \\ \mathbf{A}_{[l, 0:\lambda_{arch}[2 \times l], 0:\lambda_{arch}[2 \times (l-z)], \lfloor \frac{M_{ch}}{2} \rfloor - \lfloor \frac{\lambda_{arch}[2 \times l+1]}{2} \rfloor : \lfloor \frac{M_{ch}}{2} \rfloor + \lfloor \frac{\lambda_{arch}[2 \times l+1]}{2} \rfloor]} = 1 & , \text{otherwise} \end{cases}$$

$$(8)$$

Again, $\boldsymbol{\lambda}_{arch}[2 \times (l-z)]$ is the last entry corresponding to the non-zero input channel in $\boldsymbol{\lambda}_{arch}[2 \times l]$. Similar to the linear operation, at layer $l$, if $\boldsymbol{\lambda}_{arch}[2 \times l]$ is all zero, then the resulting $\mathbf{A}_{[l,:,:,:,:]}$ would contain only zeros and represent a "No-op" in the DOD model. Otherwise, assume we want obtain a smaller kernel size, $K_l \leq M_k$ at layer $l$, the corresponding $\mathbf{A}_{[l,:,:,:,:]}$ pads zeros around the size $M_{ch} \times k \times k$ center (See Figure B2). The masked weights $\mathbf{A}_{[l,:,:,:,:]} \odot \widehat{\mathbf{W}}_{\phi,l}$ are equivalent to obtaining smaller-size kernel weights. Notice that, when kernel sizes are different, the output of the layer's operation will also differ (smaller kernels would result in larger output size); therefore, we need to guarantee the spatial size by similarily padding zeros around the input of that convolutional layer. The padding is similar to how we construct the architecture masking $\mathbf{A}$ and similar to the padding approach discussed in Wang et al. (2020).

**Extending to Other Architectures.** We envision our proposed HN can be extended to other architectures, for example, to tune the number of attention heads and dimensions of query, key and values in the multi-headed attention mechanism Vaswani et al. (2017). We will continuously work on the implementations for other architectures, as more complex DOD models are developed recently.

# C  ADDITIONAL EXPERIMENT SETTINGS AND RESULTS

## C.1  DATASETS

To build a comprehensive testbed, we use 15 OD datasets from the DAMI repository[3] and 20 OD datasets from the ODDS repository[4]. All of these benchmark datasets are widely used in OD research. We provide the dataset summary in Table C1.

**Table C1:** The tabular testbed includes 35 datasets from DAMI and ODDS repositories.

| Dataset | Source | #Samples | #Dims | %Outlier |
|---|---|---|---|---|
| DAMI_Annthyroid | DAMI | 7129 | 21 | 7.49 |
| DAMI_Cardiotocography | DAMI | 2114 | 21 | 22.04 |
| DAMI_Glass | DAMI | 214 | 7 | 4.21 |
| DAMI_HeartDisease | DAMI | 270 | 13 | 44.44 |
| DAMI_PageBlocks | DAMI | 5393 | 10 | 9.46 |
| DAMI_PenDigits | DAMI | 9868 | 16 | 0.2 |
| DAMI_Pima | DAMI | 768 | 7 | 34.9 |
| DAMI_Shuttle | DAMI | 1013 | 9 | 1.28 |
| DAMI_SpamBase | DAMI | 4207 | 57 | 39.91 |
| DAMI_Stamps | DAMI | 340 | 9 | 9.12 |
| DAMI_Waveform | DAMI | 3443 | 21 | 2.9 |
| DAMI_WBC | DAMI | 223 | 9 | 4.48 |
| DAMI_WDBC | DAMI | 367 | 30 | 2.72 |
| DAMI_Wilt | DAMI | 4819 | 5 | 5.33 |
| DAMI_WPBC | DAMI | 198 | 33 | 23.74 |
| ODDS_annthyroid | ODDS | 7200 | 6 | 7.42 |
| ODDS_arrhythmia | ODDS | 452 | 274 | 14.6 |
| ODDS_breastw | ODDS | 683 | 9 | 34.99 |
| ODDS_glass | ODDS | 214 | 9 | 4.21 |
| ODDS_ionosphere | ODDS | 351 | 33 | 35.9 |
| ODDS_letter | ODDS | 1600 | 32 | 6.25 |
| ODDS_lympho | ODDS | 148 | 18 | 4.05 |
| ODDS_mammography | ODDS | 11183 | 6 | 2.32 |
| ODDS_mnist | ODDS | 7603 | 100 | 9.21 |
| ODDS_musk | ODDS | 3062 | 166 | 3.17 |
| ODDS_optdigits | ODDS | 5216 | 64 | 2.88 |
| ODDS_pendigits | ODDS | 6870 | 16 | 2.27 |
| ODDS_satellite | ODDS | 6435 | 36 | 31.64 |
| ODDS_satimage-2 | ODDS | 5803 | 36 | 1.22 |
| ODDS_speech | ODDS | 3686 | 400 | 1.65 |
| ODDS_thyroid | ODDS | 3772 | 6 | 2.47 |
| ODDS_vertebral | ODDS | 240 | 6 | 12.5 |
| ODDS_vowels | ODDS | 1456 | 12 | 3.43 |
| ODDS_wbc | ODDS | 378 | 30 | 5.56 |
| ODDS_wine | ODDS | 129 | 13 | 7.75 |

---

[3]DAMI repository: `https://www.dbs.ifi.lmu.de/research/outlier-evaluation/DAMI/`

[4]ODDS repository: `https://odds.cs.stonybrook.edu/`

**Table C2:** HYPER and baselines for time and performance comparison with categorization by whether it selects models (2nd column), uses meta-learning (3rd column), and requires model building at the test time (4th column). Overall, HYPER (with patience $p = 3$) achieves the best detection performances (also see Fig. 4 and 5). Compared to the SOTA ELECT, HYPER has markedly shorter offline and online time with the help of HN.

| Method | Model Selection | Meta Learning | Zero shot | Offline Time (mins.) | Average Online Time (mins.) | Median Online Time (mins.) | Avg. ROC Rank ($\downarrow$ better) |
|--------|:---:|:---:|:---:|:---:|:---:|:---:|:---:|
| Default | ✗ | ✗ | ✓ | N/A | 0 | 0 | 59.54% |
| Random | ✗ | ✗ | ✓ | N/A | 0 | 0 | 56.03% |
| MC | ✓ | ✗ | ✗ | N/A | 215 | 277 | 56.42% |
| GB | ✓ | ✓ | ✓ | 7,461 | 0 | 0 | 46.68 % |
| ISAC | ✓ | ✓ | ✓ | 7,466 | 1 | 1 | 41.81% |
| AS | ✓ | ✓ | ✓ | 7,465 | 1 | 1 | 52.22% |
| MetaOD | ✓ | ✓ | ✓ | 7,525 | 1 | 1 | 39.18% |
| ELECT | ✓ | ✓ | ✗ | 7,611 | 59 | 71 | 36.21% |
| Ours | ✓ | ✓ | ✗ | 1,320 | 14 | 17 | 29.54% |

## C.2 ALGORITHM SETTINGS AND BASELINES

In this section, we present additional experiment settings and describe the baselines. For detailed implementation information, please see `https://github.com/inreview23/HYPER`.

**Setting of the HN**: The HN utilized in the experiments consists of two hidden layers, each containing 200 neurons. It is configured with a learning rate of 1e-4, a dropout rate of 0.2, and a batch size of 512. We find this setting give enough capacity to generate various weights for linearAEs. Because of the meta-learning setting, the hyperparameters of HN can be tested with validation data and test results, on historical data.

**Meta-training for $f_{\text{val}}$.** Table C6 includes the HP search space for training. In the table, compression rate refers to how many of the widths to shrink between two adjacent layers. For example, if the first layer has width of 6, compression_rate equals 2 would gvie the next layer width equal to 3. We also notice that some datasets may have smaller numbers of features. Thus, with the corresponding compression rate, we also have discretized the width to the nearest integer number. Thus, for some datasets, the HP search space will be smaller than 240.

**HN (Re-)Training during the Online Phase**: In order to facilitate effective local re-training, we set a training epoch of $T = 100$ for each iteration, indicating the sampling of 100 local HPs for HN retraining. In Eq. (4), we designate the number of sampled HPs and the sampling factor as 500, i.e., $V_\lambda = V_\sigma = 500$. The minimum depth and maximum depth of the searched DOD models are set to 2 and 8, respectively. Essentially, we tune the DOD model depth within the integer range of 2 to 8.

**Convergence**: To achieve favorable performance within a reasonable timeframe, we set the patience value as $p = 3$ in the experiment. Further analysis for the impact of patience is available in §C.3.2.

**Baselines**: We have incorporated 8 baselines, encompassing a spectrum from simple to state-of-the-art (SOTA) approaches. Table C2 offers a comprehensive conceptual comparison of these baselines.

(*i*) *no model selection*:

(1) **Default** employs the default HPs utilized in the widely-used OD library PyOD Zhao et al. (2019). This serves as the default option for practitioners when no additional information is available.

(2) **Random** randomly selects an HP/model (the reported performance represents the expected value obtained by averaging across all DOD models).

(*ii*) *model selection without meta-learning*:

(3) **MC** Ma et al. (2023) utilizes the consensus among a group of DOD models to assess the performance of a model. A model is considered superior if its outputs are closer to the consensus of the group. MC necessitates the construction of a model group during the testing phase. For more details, please refer to a recent survey Ma et al. (2021).

(*iii*) **model selection by meta-learning** requires first building a corpus of historical datasets on a group of defined DOD models and then selecting the best from the model set at the test time. Although these baselines utilize meta-learning, none of them take advantage of the HN for acceleration.

(4) **Global Best (GB)** selects the best-performing model based on the average performance across historical datasets.

(5) **ISAC Kadioglu et al. (2010)** groups historical datasets into clusters and predicts the cluster of the test data, subsequently outputting the best model from the corresponding cluster.

(6) **ARGOSMART (AS) Nikolic et al. (2013)** measures the similarity between the test dataset and all historical datasets, and then outputs the best model from the most similar historical dataset.

(7) **MetaOD** Zhao et al. (2021) employs matrix factorization to capture both dataset similarity and model similarity, representing one of the state-of-the-art methods for unsupervised OD model selection.

(8) **ELECT** Zhao et al. (2022) iteratively identifies the best model for the test dataset based on performance similarity to the historical dataset. Unlike the above meta-learning approaches, ELECT requires model building during the testing phase to compute performance-based similarity.

**Baseline Model Set**. We use the same HP search spaces for baseline models as well as the HN-trained models. Table C6 provides the detailed HP search space.

## C.3 ADDITIONAL ABLATIONS

### C.3.1 EFFECT OF META-INITIALIZATION

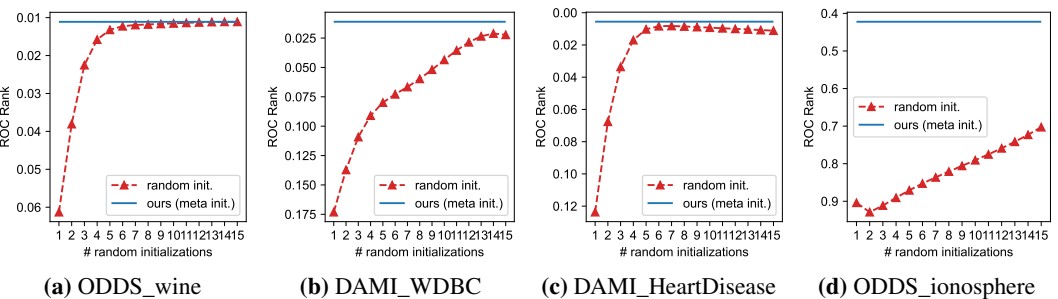

**(a)** ODDS_wine     **(b)** DAMI_WDBC     **(c)** DAMI_HeartDisease     **(d)** ODDS_ionosphere

**Figure C3:** Comparison of ROC Rank (lower is better) of HYPER with meta-initialization (in blue) with increasing numbers of randomly initialized HNs. For instance, it needs 9 randomly initialized HNs to achieve the same performance as HYPER on ODDS_wine. In general, HYPER shows better efficiency in finding a good model with much less running time.

As mentioned in §3.2, HYPER initializes the HPs to the "global best HPs" derived from the historical training datasets. Specifically, in each fold comprising 7 test datasets, we utilize the HPs that yield the best average performance across the remaining 28 training datasets as the initial HPs. This meta-initialization approach leverages meta-learning to initialize HYPER with a potentially promising HP configuration.

In Figure C3, we demonstrate the effectiveness of meta-initialization by comparing it with random initialization on five datasets. In addition to utilizing meta-initialization, one could run HYPER multiple times with randomly initialized HPs and select the best model based on $f_{val}$. However, it should be noted that $f_{val}$ serves as a proxy validator rather than ground truth. Therefore, including more randomly initialized HNs does not guarantee a monotonic improvement in the selected model's performance. Nonetheless, increasing the number of random initializations is likely to yield a better performance by exploring a broader range of search spaces.

To simulate this scenario, we vary the number of random initializations (x-axis) and record all the $f_{val}$ values along with the corresponding ROC Rank. For each dataset, we select the best model based on $f_{val}$ across *all* trials. We increase the number of random trials from 1 to 15, where the highest $f_{val}$ value among the 15 random initialized trials is chosen as the best model. The y-axis represents the average performance from independent trials.

The figure clearly demonstrates the advantage of meta-initialization as a strong starting point for HYPER's HP tuning. For example, on the `ODDS_wine` dataset (refer to Fig. C3a), it requires 9 randomly initialized HNs to attain the same performance as our approach with meta-initialization, showing a 9-fold increase in the time required for online selection. In other cases (Fig. C3b, C3c, and C3d), training 15 randomly initialized HNs fails to achieve the same performance as meta-initialization, further validating its advantages.

### C.3.2   EFFECT OF PATIENCE

As described in §3.1, the convergence criterion for HYPER is based on the highest predicted performance by $f_{\text{val}}$ remaining unchanged for $\rho$ consecutive iterations. A larger value of $\rho$, also known as "patience", requires more iterations for convergence but is likely to yield better results. As illustrated in Fig. C4, increasing the value of $\rho$ allows for more exploration and potentially better performance. However, this also prolongs the convergence time.

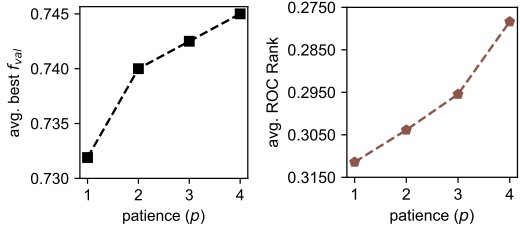

**Figure C4:** Analysis of the effect of patience $\rho$: (left) avg. $f_{\text{val}}$ value change when increasing $\rho$ from 1 to 4; (right) avg. ROC Rank with increasing $\rho$. Larger $\rho$ leads to more exploration and likely better performance.

In our experiments, we set $\rho=3$ to balance performance and runtime. The specific value of $\rho$ can be determined through cross-validation over the historical datasets, taking into account the specific criteria and requirements of the underlying application.

### C.4   ADDITIONAL RESULTS

In addition to the distribution plot in Fig. 5, we provide the $p$-values of Wilcoxon signed rank test between HYPER and baselines in C3. See §5 for the experiment analysis.

**Table C3:** Pairwise statistical tests between HYPER and baselines by Wilcoxon signed rank test. HYPER are statistically better than the baselines and its variants at different significance levels. See Fig.5 for distribution rank with statistical comparison.

| Ours | Baseline | p-value |
|------|----------|---------|
| **Ours** | Default | 0.0016 |
| **Ours** | Random | 0.0001 |
| **Ours** | ISAC | 0.0025 |
| **Ours** | AS | 0.0052 |
| **Ours** | MetaOD | 0.0030 |
| **Ours** | Global Best | 0.0015 |
| **Ours** | MC | 0.0005 |
| **Ours** | ELECT | 0.0568 |
| **Ours** | Ours (reg&width) | 0.0001 |
| **Ours** | Ours (reg&depth) | 0.0001 |
| **Ours** | Ours (reg only) | 0.0001 |

We present the full results in Table C4 and C5. Note that ROC Rank is based on each method's performance in regard to the baseline model set's performance described in §C.2.

**Table C4:** ROC of the evaluated methods. The best method per dataset (row) is highlighted in bold.

| Dataset | Default | Random | MC | GB | ISAC | AS | MetaOD | ELECT | Ours |
|---|---|---|---|---|---|---|---|---|---|
| DAMI_Annthyroid | **0.7124** | 0.5972 | 0.6123 | 0.5929 | 0.6018 | 0.5873 | 0.6050 | 0.6148 | 0.5888 |
| DAMI_Cardiotocography | 0.7159 | 0.7202 | 0.7024 | 0.7740 | 0.7571 | 0.6940 | 0.6458 | 0.7818 | **0.7866** |
| DAMI_Glass | **0.7442** | 0.7055 | 0.6304 | 0.7244 | 0.6699 | 0.7230 | 0.7225 | 0.7431 | 0.6917 |
| DAMI_HeartDisease | 0.3045 | 0.4276 | 0.4214 | 0.5250 | 0.5382 | 0.4214 | 0.5312 | 0.5348 | **0.5926** |
| DAMI_PageBlocks | 0.8722 | 0.9107 | 0.9219 | 0.9162 | **0.9255** | 0.9002 | 0.6247 | 0.8791 | 0.9215 |
| DAMI_PenDigits | 0.3837 | 0.5248 | 0.5422 | 0.5491 | 0.5069 | **0.6953** | 0.6278 | 0.5084 | 0.6792 |
| DAMI_Pima | 0.4005 | 0.5508 | 0.4855 | 0.6216 | 0.6279 | 0.6158 | 0.4862 | 0.6281 | **0.6595** |
| DAMI_Shuttle | 0.6453 | 0.9462 | 0.9400 | 0.9342 | 0.9436 | **0.9530** | 0.5525 | 0.9405 | 0.9391 |
| DAMI_SpamBase | 0.5208 | 0.5232 | 0.4907 | 0.5210 | 0.5263 | **0.5552** | 0.5307 | 0.5135 | 0.5525 |
| DAMI_Stamps | 0.8687 | 0.8687 | 0.8926 | 0.8981 | **0.9079** | 0.8618 | 0.7112 | 0.8897 | 0.9003 |
| DAMI_Waveform | 0.6810 | 0.6772 | 0.6560 | 0.6941 | 0.6924 | 0.6890 | 0.6900 | **0.7019** | 0.6929 |
| DAMI_WBC | 0.7493 | 0.9769 | 0.9770 | 0.9682 | 0.9742 | 0.9779 | 0.9809 | 0.9779 | **0.9826** |
| DAMI_WDBC | 0.8092 | 0.8366 | 0.8146 | 0.8597 | 0.8683 | 0.8092 | 0.8361 | 0.8213 | **0.9039** |
| DAMI_Wilt | **0.5080** | 0.4524 | 0.4832 | 0.4653 | 0.4700 | 0.4700 | 0.4714 | 0.4700 | 0.3709 |
| DAMI_WPBC | 0.4090 | 0.4464 | 0.3972 | 0.4679 | 0.4548 | 0.4285 | 0.4456 | 0.4726 | **0.4824** |
| ODDS_annthyroid | **0.7353** | 0.6981 | 0.6963 | 0.6982 | 0.7067 | 0.7067 | 0.6903 | 0.7058 | 0.7014 |
| ODDS_arrhythmia | 0.7769 | 0.7786 | 0.7810 | 0.7767 | **0.7831** | 0.7798 | 0.7824 | 0.7807 | 0.7827 |
| ODDS_breastw | 0.5437 | 0.6187 | 0.8939 | **0.9071** | 0.8032 | 0.7986 | 0.5913 | 0.8649 | 0.9045 |
| ODDS_glass | **0.6195** | 0.5849 | 0.5453 | 0.5897 | 0.5962 | 0.5962 | 0.5654 | 0.5957 | 0.5993 |
| ODDS_ionosphere | 0.8708 | 0.8497 | **0.8711** | 0.8252 | 0.8422 | 0.8350 | 0.8727 | 0.8686 | 0.8509 |
| ODDS_letter | 0.5555 | 0.5758 | 0.5918 | 0.6068 | 0.6244 | 0.6155 | **0.6446** | 0.6211 | 0.6102 |
| ODDS_lympho | 0.9096 | 0.9959 | 0.9988 | 0.9842 | 0.9929 | 0.9953 | 0.9971 | **1.0000** | 0.9925 |
| ODDS_mammography | 0.5287 | 0.7612 | 0.7233 | 0.8362 | 0.7189 | 0.7116 | **0.8640** | 0.7673 | 0.8542 |
| ODDS_mnist | 0.8518 | 0.8915 | 0.8662 | 0.8959 | 0.9011 | 0.8580 | **0.9070** | 0.9032 | 0.8994 |
| ODDS_musk | 0.9940 | **1.0000** | **1.0000** | **1.0000** | **1.0000** | **1.0000** | **1.0000** | **1.0000** | **1.0000** |
| ODDS_optdigits | 0.5104 | 0.4950 | 0.5092 | 0.4806 | 0.5115 | 0.5171 | 0.4973 | 0.5338 | **0.5584** |
| ODDS_pendigits | 0.9263 | 0.9295 | 0.9265 | 0.9305 | 0.9208 | 0.9386 | 0.9360 | 0.9346 | **0.9435** |
| ODDS_satellite | **0.7681** | 0.7284 | 0.7445 | 0.7352 | 0.7433 | 0.7324 | 0.7486 | 0.7571 | 0.7432 |
| ODDS_satimage-2 | 0.9707 | 0.9826 | **0.9865** | 0.9744 | 0.9838 | 0.9798 | 0.9871 | 0.9786 | 0.9853 |
| ODDS_speech | 0.4761 | 0.4756 | 0.4692 | 0.4726 | **0.4832** | 0.4692 | 0.4706 | 0.4774 | 0.4707 |
| ODDS_thyroid | **0.9835** | 0.9661 | 0.9652 | 0.9535 | 0.9635 | 0.9652 | 0.9740 | 0.9689 | 0.9667 |
| ODDS_vertebral | **0.6019** | 0.5378 | 0.5629 | 0.5253 | 0.4602 | 0.5629 | 0.4657 | 0.5629 | 0.4757 |
| ODDS_vowels | 0.4897 | 0.5903 | 0.5965 | 0.6309 | 0.6414 | **0.6686** | 0.6216 | 0.5247 | **0.6686** |
| ODDS_wbc | 0.4146 | 0.8401 | 0.7640 | 0.8808 | 0.8745 | 0.8469 | 0.8770 | 0.8469 | **0.9289** |
| ODDS_wine | 0.7864 | 0.5430 | 0.4084 | 0.7539 | 0.5387 | 0.4084 | 0.6296 | 0.6218 | **0.8287** |

**Table C5:** ROC Rank (smaller the better) of the evaluated methods. The best method per dataset (row) is highlighted in bold.

| Dataset | Default | Random | MC | GB | ISAC | AS | MetaOD | ELECT | Ours |
|---|---|---|---|---|---|---|---|---|---|
| **DAMI_Annthyroid** | **0.0079** | 0.5317 | 0.1111 | 0.6031 | 0.4206 | 0.7143 | 0.3492 | 0.0794 | 0.6984 |
| **DAMI_Cardiotocography** | 0.5635 | 0.4683 | 0.6190 | 0.0476 | 0.1984 | 0.6825 | 1.0000 | 0.0476 | **0.0238** |
| **DAMI_Glass** | **0.0556** | 0.7222 | 0.9365 | 0.1984 | 0.9048 | 0.5556 | 0.5635 | 0.0595 | 0.8016 |
| **DAMI_HeartDisease** | 0.9841 | 0.5556 | 0.5873 | 0.1032 | 0.0238 | 0.5714 | 0.0476 | 0.0317 | **0.0079** |
| **DAMI_PageBlocks** | 1.0000 | 0.4921 | 0.2698 | 0.4048 | **0.1667** | 0.7937 | 1.0000 | 0.9841 | 0.2937 |
| **DAMI_PenDigits** | 0.9048 | 0.5317 | 0.4722 | 0.4206 | 0.6429 | **0.0714** | 0.1349 | 0.6270 | 0.0794 |
| **DAMI_Pima** | 1.0000 | 0.5159 | 0.9603 | 0.1508 | 0.0635 | 0.2540 | 0.6508 | 0.0556 | **0.0079** |
| **DAMI_Shuttle** | 1.0000 | 0.4444 | 0.9365 | 1.0000 | 0.5635 | **0.2381** | 1.0000 | 0.7302 | 0.9683 |
| **DAMI_SpamBase** | 0.6349 | 0.5556 | 0.9921 | 0.6270 | 0.3571 | **0.0397** | 0.1825 | 0.8016 | 0.0556 |
| **DAMI_Stamps** | 0.5238 | 0.5238 | 0.2619 | 0.1587 | **0.0635** | 0.6032 | 1.0000 | 0.2857 | 0.1349 |
| **DAMI_Waveform** | 0.4762 | 0.5556 | 0.8175 | 0.1429 | 0.2143 | 0.2619 | 0.2619 | **0.0556** | 0.1905 |
| **DAMI_WBC** | 1.0000 | 0.7381 | 0.7381 | 0.9921 | 0.9365 | 0.5873 | 0.0079 | 0.5873 | **0.0079** |
| **DAMI_WDBC** | 0.8095 | 0.5397 | 0.7619 | 0.2381 | 0.0873 | 0.8016 | 0.5556 | 0.6984 | **0.0159** |
| **DAMI_Wilt** | **0.0159** | 0.7302 | 0.0238 | 0.7143 | 0.7143 | 0.7143 | 0.0397 | 0.7143 | 0.9762 |
| **DAMI_WPBC** | 1.0000 | 0.4365 | 1.0000 | 0.2302 | 0.3413 | 0.8175 | 0.4524 | 0.1667 | **0.0079** |
| **ODDS_annthyroid** | **0.0397** | 0.3651 | 0.3889 | 0.3571 | 0.2381 | 0.2302 | 0.7937 | 0.2540 | 0.3016 |
| **ODDS_arrhythmia** | 0.7063 | 0.5556 | 0.2302 | 0.7222 | **0.1270** | 0.3889 | 0.1587 | 0.2857 | 0.1349 |
| **ODDS_breastw** | 0.5794 | 0.5317 | 0.1032 | 0.0714 | 0.3095 | 0.3333 | 0.5317 | 0.1905 | **0.0794** |
| **ODDS_glass** | **0.2143** | 0.3571 | 0.9048 | 0.3333 | 0.2937 | 0.2857 | 0.5159 | 0.3095 | 0.2857 |
| **ODDS_ionosphere** | 0.0872 | 0.4841 | **0.0873** | 0.9683 | 0.5714 | 0.8254 | 0.0476 | 0.1587 | 0.4603 |
| **ODDS_letter** | 0.6746 | 0.6429 | 0.4762 | 0.3333 | **0.0476** | 0.1349 | 0.0079 | 0.0635 | 0.2539 |
| **ODDS_lympho** | 1.0000 | 0.4921 | 0.2143 | 1.0000 | 0.9365 | 0.6429 | 0.4365 | **0.0437** | 0.9365 |
| **ODDS_mammography** | 1.0000 | 0.3889 | 0.8968 | 0.2381 | 0.9127 | 0.9286 | **0.0317** | 0.3571 | 0.0714 |
| **ODDS_mnist** | 0.9286 | 0.6746 | 0.8651 | 0.6190 | 0.4603 | 0.9048 | **0.1270** | 0.3492 | 0.5238 |
| **ODDS_musk** | 1.0000 | 0.8810 | **0.3730** | **0.3730** | **0.3730** | **0.3730** | **0.3730** | **0.3730** | **0.3730** |
| **ODDS_optdigits** | 0.3651 | 0.5714 | 0.4048 | 0.6825 | 0.3333 | 0.2063 | 0.5238 | 0.1032 | **0.0397** |
| **ODDS_pendigits** | 0.7222 | 0.6587 | 0.7222 | 0.6269 | 0.8254 | 0.1825 | 0.2619 | 0.3730 | **0.0397** |
| **ODDS_satellite** | **0.0238** | 0.7222 | 0.2778 | 0.6825 | 0.3651 | 0.7063 | 0.0873 | 0.0317 | 0.3889 |
| **ODDS_satimage-2** | 1.0000 | 0.6270 | 0.0556 | 0.9841 | 0.4048 | 0.8413 | **0.0317** | 0.9048 | 0.0714 |
| **ODDS_speech** | 0.4048 | 0.4286 | 0.8651 | 0.5873 | **0.1825** | 0.8730 | 0.6825 | 0.3810 | 0.6746 |
| **ODDS_thyroid** | **0.0079** | 0.4841 | 0.5556 | 1.0000 | 0.6905 | 0.5635 | 0.0476 | 0.4603 | 0.4762 |
| **ODDS_vertebral** | **0.0159** | 0.6984 | 0.3571 | 0.7381 | 0.9127 | 0.5159 | 0.8730 | 0.3571 | 0.8254 |
| **ODDS_vowels** | 1.0000 | 0.5873 | 0.5397 | 0.3254 | 0.2063 | **0.1111** | 0.3730 | 0.8016 | 0.1111 |
| **ODDS_wbc** | 1.0000 | 0.6349 | 0.9762 | 0.1746 | 0.2460 | 0.5714 | 0.2063 | 0.5714 | **0.0079** |
| **ODDS_wine** | 0.0238 | 0.4841 | 0.9603 | 0.0397 | 0.5000 | 0.9603 | 0.3333 | 0.3810 | **0.0159** |
| **Avg. ROC Rank** | 0.5954 | 0.5603 | 0.5642 | 0.4668 | 0.4181 | 0.5222 | 0.3918 | 0.3621 | 0.2954 |

**Table C6:** Hyperparameter search space for both free-range and HN models. We give the list of HPs as well as the range of the selected HPs.

| List of Hyperparameters (HPs) | # HPs |
|---|---|
| n_layers: [2,4,6,8] | 4 |
| compression_rate: [1.0,1.2,1.4,1.6,1.8,2.0,2.2,2.4,2.6,2.8,3.0] | 10 |
| dropout: [0.0,0.2,0.4] | 3 |
| weight_decay: [0.0,1e-6,1e-5] | 3 |
| Total Number: | 240 |