# OpenReview forum: "Fast Unsupervised Deep Outlier Model Selection with Hypernetworks"
_ICLR.cc/2024/Conference — ICLR 2024 Conference Withdrawn Submission_

### Official Review · Reviewer_LDgS · 2023-10-24

**Soundness:** 2 fair
**Presentation:** 1 poor
**Contribution:** 2 fair
**Rating:** 3
**Confidence:** 3

**Summary:**

In this study, the authors introduce 'Hyper,' a method for hyperparameter-tuning in DOD models. 'Hyper' aims to solve two key issues: unsupervised validation and the efficient exploration of HP/model spaces. This method is inspired by the concept of a hypernetwork (HN) capable of mapping hyperparameters to the optimal weights of a primary DOD model. Uniquely, 'Hyper' leverages a single HN designed to dynamically produce weights for DOD models.

**Strengths:**

1. A large set of experiments was conducted, involving comparisons with 8 baselines across 35 datasets.

**Weaknesses:**

1. Utilizing the test set during training is not a valid practice in machine learning.
2. The majority of the datasets and networks (3-layer convolutional networks) employed in the study are relatively small, raising questions about the scalability and practicality of the proposed method when applied to larger datasets.
3. The presentation and structure of this paper significantly hinder its readability, to the point where I find it challenging to proceed with the review. I recommend that the authors make significant changes before resubmitting.

     1） Many sentences are challenging to comprehend due to their complexity or lack of context.

     2） Related work within the Introduction is unconventional and may confuse readers.

     3）The citation style also needs to be standardized, as references are missing parentheses.

     4）The frequent use of abbreviations can impede understanding.

     5）The inclusion of the main paper's copy in the Supplementary Material is redundant.

**Questions:**

Please refer to the weakness.

---

### Official Review · Reviewer_Th7t · 2023-10-31

**Soundness:** 2 fair
**Presentation:** 2 fair
**Contribution:** 2 fair
**Rating:** 5
**Confidence:** 3

**Summary:**

The paper proposes a new method for unsupervised outlier detection model selection (UODMS). UODMS has two key challenges: (1) validation without supervision and (2) efficient search of the HP/model space. To tackle the unsupervised validation, they follow the existing work to learn a model performance estimator with meta-learning. Instead of training individual deep OD models when searching the HPs, they propose to learn hypernetworks that map HPs to optimal deep model parameters, and therefore speed up the search. The proposed UODMS framework consists of two stages. On the off-line meta training, a single global hypernetwork and the performance estimator are trained on historical datasets. Given a new dataset, the proposed method alternates between updating the local hypernetwork around the current hyperparameters and optimizing the current hyperparameters. The authors provide extensive experiments to show that the proposed method improves UODMS performance and speeds up the running of the algorithm.

**Strengths:**

1. The paper studies a well-motivated and important problem. In outlier detection, the labeled outliers are often not available. The performance of deep outlier detection models usually depends on the selected hyperparameters. UDOMS is an important problem in practice.

2. The paper provides an extensive empirical evaluation to showcase the efficacy of the proposed method. The proposed method achieves the lowest average ROC ranking on 35 tabular datasets.

**Weaknesses:**

1. In comparison to existing works, e.g., MetaOD, the major contribution of this paper is learning hypernetworks for saving individual model training time. From Fig. 4 we can see that MetaOD is actually faster than the proposed method. The training of hypernetworks becomes more complicated and involves new hyperparameters.

2. The whole framework involves several learnable components, e.g., data embedding, model embedding, performance estimator, and the hypernetworks. The final UDOMS performance depends on the quality of all these trained components. However, the paper does not provide an ablation study to demonstrate the impact of each component.

**Questions:**

1. The hypernetworks map hyperparameters to high-dimensional model parameters. I wonder how stable is the training of the HNs. The training of HNs also depends on model embedding and data embedding. What if the model embedding and data embedding are not good enough?

2. On the online stage, the alternative optimization seems also sensitive to the initial HNs and the exploration strength of HPs. If the goal of UDOMS is to have an automatic tunning of HPs, how do you select the hyperparameters related to HNs and the alternative optimization on the online stage?

3. Could the authors also provide the average AUCs besides the average ranking?

---

### Official Review · Reviewer_7fKL · 2023-11-01

**Soundness:** 2 fair
**Presentation:** 2 fair
**Contribution:** 3 good
**Rating:** 5
**Confidence:** 3

**Summary:**

In the paper authors propose Hypernetwork framework for deep Outlier detection models. New method is dedicated two main task 1. validation without supervision 2. efficient search of the hyperparameters tiuning.

**Strengths:**

1. The paper introduces an interesting setting.
2. Method gives nice numerical results.

**Weaknesses:**

1. Notation HP and HN is misleading. Maybe is better to use full names.
2. It is hard to understand the setting of the model. In OD we do not have labels. But in meta training on historical data, we use labels.
3. Figure 2 is not clear, especially part 3.2
4. Experiments work with fully connected AutoEncoder (AE) for DOD on tabular data.
The architecture is simple, and the datasets are also.
5. Authors should use image datasets.
6. Authors should present results on convolution layers.
7. Authors claim: The sensitivity of outlier detectors to hyperparameter (HP) choices is well studied Campos et al.(2016a). But the citation is quite old. New OD methods are also sensitive to HPs?
8. It is not obvious how the hypernetwork-based solution is time and memory-consuming.

**Questions:**

1. How does the model work with large dimensional data like images?
2. How does the model work with convolutional layers?
3. What is the difference in time training?
4. Is it possible to use the model for classic classification tasks?

---

### Official Review · Reviewer_2NEK · 2023-11-01

**Soundness:** 2 fair
**Presentation:** 3 good
**Contribution:** 2 fair
**Rating:** 5
**Confidence:** 3

**Summary:**

This paper proposes a new framework for unsupervised deep outlier model selection using hypernetworks. HYPER is a novel approach to effectively tune hyperparameters and select models for deep neural network based outlier detection (DOD). The authors tackle key challenges in unsupervised DOD, including validation without supervision and efficient search of the hyperparameter/model space. HYPER uses a hypernetwork to dynamically generate weights for many DOD models, which are then trained on the same data. The hypernetwork is trained to output weights that maximize the performance of the ensemble of models on a validation set. This allows HYPER to effectively search the hyperparameter/model space and select the best model for a given dataset. The authors conducted extensive experiments on benchmark datasets and compared HYPER to several state-of-the-art methods. They found that HYPER achieved significant performance improvements and speed-up in OD tasks.  Overall, HYPER is a promising approach to unsupervised DOD that addresses key challenges and achieves state-of-the-art performance.

**Strengths:**

1. The paper introduces a novel approach to effectively tune hyperparameters and select models for deep neural network based outlier detection. The use of a hypernetwork to dynamically generate weights for many DOD models is a unique and innovative idea.

2. The authors tackle key challenges in unsupervised DOD, including validation without supervision and efficient search of the hyperparameter/model space. By addressing these challenges, HYPER achieves significant performance improvements and speed-up in OD tasks.

3. The authors conducted extensive experiments on benchmark datasets and compared HYPER to several state-of-the-art methods. The results show that HYPER outperforms other methods in terms of both accuracy and efficiency. They also show that their framework is able to find models with statistically better detection performance than the default HPs.

**Weaknesses:**

1. The authors do not compare their framework to methods that use labeled data. It is unclear how their framework would perform on datasets with labeled data, which is the setting in which outlier detection is typically applied.

2. The authors only evaluate their framework on a relatively small set of benchmark datasets. It is possible that their framework would not generalize well to other datasets. While the authors demonstrate the effectiveness of HYPER on benchmark datasets, there is no evaluation on real-world applications. It is unclear how well HYPER would perform on more complex and diverse datasets.

3. The authors do not provide any theoretical guarantees on the performance of their framework. It is unclear how well their framework would perform on datasets with different characteristics.

4. The authors' framework is computationally expensive to train. This could be a limitation for other researchers who need to select models for a large number of datasets to use in practice.

**Questions:**

1. Is there any detailed analysis of the hypernetwork architecture used in HYPER? It would be useful to understand how different hypernetwork architectures affect the performance of HYPER.

2. What are the models and legend in Figure 3, why losses increase in the first hundreds of epochs? This figure is hard to understand.

3. In Figure 5, can you explain the setup of the bottom three?

4. Since the values over iterations in Figure 6 doesn't appear to plateau, how to use HP schedules to accommodate a larger model space?

---

### Official Review · Reviewer_izNE · 2023-11-04

**Soundness:** 2 fair
**Presentation:** 1 poor
**Contribution:** 2 fair
**Rating:** 3
**Confidence:** 4

**Summary:**

The authors propose a meta-learning approach for selecting models and hyperparameters for outlier detection. Their problem setup assumes that there are no outlier labels during the test, therefore it relies on a "detection performance predictor" that is trained using auxiliary tasks. This predictor depends on the hyperparameters, the data, and the outlier scores of the selected model conditioned to the selected hyperparameters. The outlier scores are generated by the model using parameters generated by a hypernetwork. The only input to the hypernetwork is the hyperparameters. Thus the hypernetwork outputs the weights of the model after training it under a certain hyperparameter setup. During the test, they use iteratively the meta-learned performance predictor to query hyperparameters and to retrain the hypernetwork. The authors perform evaluations in 35 datasets, demonstrating an improvement in performance and speed.

**Strengths:**

-  The problem setup is realistic and important, as there are outliers in real-data scenarios. Moreover, its detection helps to improve the modeling.
-  The approach seems to improve performance and inference speed.

**Weaknesses:**

- The manuscript is very difficult to read and it is confusing in some parts, especially when explaining the proposed method.
- Due to the lack of effective explanation, the authors consume 7 out of 9 pages introducing the method and just dedicate 2 pages to the experiments. Given that this is a paper with empirical methods, it should dedicate more space elaborating on experiments and results.
- There is no clear explanation of the meta-learning algorithm, which is a very important stage in this paper.
- Although the method seems to work, it appears unnecessarily complicated. For instance, it learns a performance predictor that is used on the new dataset. Since it is not possible to measure the performance on this dataset due to the lack of labels, they use the performance predictor as the function to optimize. However, this function is a white-box function, which can be optimized directly or by other more efficient methods. Thus, the Algorithm 1 could be much simpler.
- Following the point above, a very important baseline is to optimize directly the detection performance using the predictor (e.g. using gradient descent).
- It is not clear how big the hyperparameter search space actually is. Table C6 in the appendix shows only 4 hps, with 4 values. If this is the case, it is a extremely small search space.
- There are only experiments with tabular data. The question remains whether it works on other types of modalities.
- The method introduces a lot of design choices such as the hypernetwork design, the performance prediction, the learning rate for training, etc. However, there is an ablation study that does not cover these aspects.

**Questions:**

- How good is the metalearned performance predictor (final loss value)? How can we be sure this predictor is generalizing?
- Did you try other types of embeddings for the tabular data? Why did you choose the feature hashing?
- How does the performance of the method change when trading off exploitation and exploration with $\tau$?